# P2RY14 cAMP signaling regulates Schwann cell precursor self-renewal, proliferation, and nerve tumor initiation in a mouse model of neurofibromatosis

**Jennifer Patritti Cram[1,2], Jianqiang Wu[1,3], Robert A Coover[1†], Tilat A Rizvi[1], Katherine E Chaney[1], Ramya Ravindran[4], Jose A Cancelas[1,5], Robert J Spinner[6], Nancy Ratner[1,3]***

[1]Division of Experimental Hematology and Cancer Biology, Cancer & Blood Diseases Institute, Cincinnati Children's Hospital Medical Center, Cincinnati, United States; [2]Neuroscience Graduate Program, University of Cincinnati College of Medicine, Cincinnati, United States; [3]Department of Pediatrics, University of Cincinnati College of Medicine, Cincinnati, United States; [4]Molecular and Developmental Biology, Cincinnati Children's Hospital, Cincinnati, United States; [5]Hoxworth Blood Center, College of Medicine, University of Cincinnati, Cincinnati, United States; [6]Department of Neurosurgery, Mayo Clinic, Rochester, United States

**\*For correspondence:**
nancy.ratner@cchmc.org

**Present address:** [†]Dept. of Basic Pharmaceutical Sciences, High Point University, High Point, United States

**Competing interest:** The authors declare that no competing interests exist.

**Abstract** Neurofibromatosis type 1 (NF1) is characterized by nerve tumors called neurofibromas, in which Schwann cells (SCs) show deregulated RAS signaling. NF1 is also implicated in regulation of cAMP. We identified the G-protein-coupled receptor (GPCR) P2ry14 in human neurofibromas, neurofibroma-derived SC precursors (SCPs), mature SCs, and mouse SCPs. Mouse *Nf1-/-* SCP self-renewal was reduced by genetic or pharmacological inhibition of P2ry14. In a mouse model of NF1, genetic deletion of P2ry14 rescued low cAMP signaling, increased mouse survival, delayed neurofibroma initiation, and improved SC Remak bundles. P2ry14 signals via $G_i$ to increase intracellular cAMP, implicating P2ry14 as a key upstream regulator of cAMP. We found that elevation of cAMP by either blocking the degradation of cAMP or by using a P2ry14 inhibitor diminished *NF1-/-* SCP self-renewal in vitro and neurofibroma SC proliferation in in vivo. These studies identify P2ry14 as a critical regulator of SCP self-renewal, SC proliferation, and neurofibroma initiation.

## Editor's evaluation

This study explores a role for the purinergic receptor P2RY14 and cAMP signaling in Schwann cell precursor self-renewal and neurofibroma development. Importantly, the authors show that genetic and chemical inhibition of P2RY14 inhibits Schwann cell precursor self-renewal in vitro and suppresses neurofibroma development in vivo. The authors also report that these effects are mediated by an increase in cAMP signaling.

## Introduction

Neurofibromatosis type 1 (NF1) is an autosomal dominant disease that affects up to 1:2000 individuals worldwide (*Kallionpää et al., 2018*). To date, there is no cure for NF1, which is characterized by multiple, variable, clinical manifestations (*Friedman, 1998*; *Tabata et al., 2020*). At least half of the children with NF1 develop plexiform neurofibromas (PNs), which are tumors within peripheral

nerves. PN may be present at birth and show most rapid growth during the first decade of a child's life (*Nguyen et al., 2012*). PNs can occur in any cranial or peripheral nerve and have the potential to transform into lethal malignant peripheral nerve sheath tumors (MPNST) (*Prudner et al., 2020*). Neurofibroma infiltration of normal nerves in NF1 patients results in a complicated risk profile because it can cause nerve damage and compress nearby vital organs (*Kim et al., 2017*). Therefore, understanding how neurofibromas form and how to treat them is under intense investigation.

Peripheral nerve glial cells, Schwann cells (SCs), are the only cell type in neurofibromas that shows bi-allelic loss-of-function mutations in the *NF1* tumor suppressor gene (*Serra et al., 1997*; *Serra et al., 2001*). Neurofibroma SCs also show aberrant properties ex vivo, consistent with it being the primary pathogenic cell type in neurofibromas (*Sheela et al., 1990*; *Kim et al., 1995*). In mice, neural crest cells develop into SC precursors (SCPs) between embryonic day 11 (E11) and E13 (*Jessen and Mirsky, 2019*). SCPs or related boundary cap cells can serve as cells-of-origin for neurofibromas, as loss of *Nf1* in these cells causes PN formation (*Zhu et al., 2002*; *Wu et al., 2008*; *Chen et al., 2014*; *Chen et al., 2019*).

In vitro, embryonic SCPs retain multi-lineage differentiation potential and self-renewal capabilities for several passages, indicating that they are progenitor-like cells (*Jessen and Mirsky, 2019*). Mouse SCPs also express epidermal growth factor receptor⁺ (EGFR⁺) and respond to EGF with limited self-renewal (*Williams et al., 2008*). EGFR⁺ cells that co-express the SC marker S100 account for about 1.8% human neurofibroma cells (*DeClue et al., 2000*). The idea that these cells may be tumor-initiating cells is consistent with the finding that human neurofibromas sorted for co-expression of the SC marker p75⁺ and EGFR show limited self-renewal in vitro. Also, EGFR-dependent *Nf1⁻/⁻* SCPs show increased self-renewal and form neurofibromas upon transplantation (*Joseph et al., 2008*; *Williams et al., 2008*). Together, these studies suggest the presence of progenitor-like cells in neurofibromas, which depend on EGFR for self-renewal. EGFR signaling may also play additional roles in transformed SCs, as MPNST cells re-acquire EGFR expression that is absent in mature SC (*DeClue et al., 2000*).

SCPs differentiate into SCs. When associated with a single large-diameter axonal segment SCs form myelin. SCs associated with multiple smaller diameter axons become non-myelinating Remak cells (*Mirsky et al., 2008*). Neurofibroma SCs show a dramatic change in Remak SC morphology, bundling one or two few axons (*Erlandson and Woodruff, 1982*; *Zheng et al., 2008*), rather than up to 20 small diameter axons in WT nerve Remak bundles (*Harty and Monk, 2017*). Notably, while neurofibromas rarely form, elevation of EGFR in WT SCPs and SCs is sufficient to mimic this nerve disruption phenotype (*Ling et al., 2005*).

The *NF1* gene encodes neurofibromin, a GTPase activating protein that accelerates the hydrolysis of RAS-GTP to its inactive GDP-bound form downstream of EGFR (*Simanshu et al., 2017*). In SCs, loss of neurofibromin causes increases in GTP-bound RAS (*Kim et al., 1995*; *Sherman et al., 2000*), and RAS-GTP stimulates the mitogen-activated protein kinase (MAPK) pathway and other downstream pathways, including deregulation of the PI3K-AKT signaling (*Dasgupta et al., 2005*; *Johannessen et al., 2008*; *Patmore et al., 2012*; *Keng et al., 2012*). Loss of *NF1* also causes reduced levels of cyclic AMP (cAMP) in *Nf1* mutant mouse, *fly* and zebrafish (*Hegedus et al., 2007*; *Tong et al., 2002*; *Wolman et al., 2014*; *Anastasaki and Gutmann, 2014*). Whether cAMP deregulation occurs downstream of increased RAS-GTP is unclear. Neurofibromin shares homology with the yeast proteins Ira1 and Ira2, which are inhibitory regulators of the RAS-cAMP adenylyl cyclase pathway, but no evidence shows a direct role for NF1 or IRA proteins in direct regulation of cAMP in mammals (*Ballester et al., 1990*; *Martin et al., 1990*; *Xu et al., 1990*). It is unclear how, or if, regulation of cAMP is relevant to neurofibroma initiation or growth but reducing cAMP drove formation of brain tumors in cells lacking *Nf1* (*Warrington et al., 2010*).

We sought to identify molecules that might affect neurofibroma development in the SC lineage. We identified *P2ry14* as a G-protein-coupled receptor (GPCR) expressed in neurofibroma SCPs. *P2ry14* is activated by extracellular UDP and UDP-sugars and signal through G$_i$ to inhibit adenylate cyclase (AC), decreasing cAMP (*Abbracchio and Ceruti, 2006*; *Conroy et al., 2016*). Intriguingly, *P2ry14* regulates homeostasis of hematopoietic stem/progenitor cells (*Cho et al., 2014*). Also, satellite glial cells and SCs have been reported to express *P2ry14* in vitro (*Patritti-Cram et al., 2021*). Here, we show that, in vitro, *P2ry14* inhibition decreases mouse SCP self-renewal by modulation of cAMP. In vivo, *P2ry14* knockout increased mouse survival, decreased SC proliferation, improved nerve Remak bundle morphology, and decreased tumor initiation. Pharmacological elevation of cAMP diminished

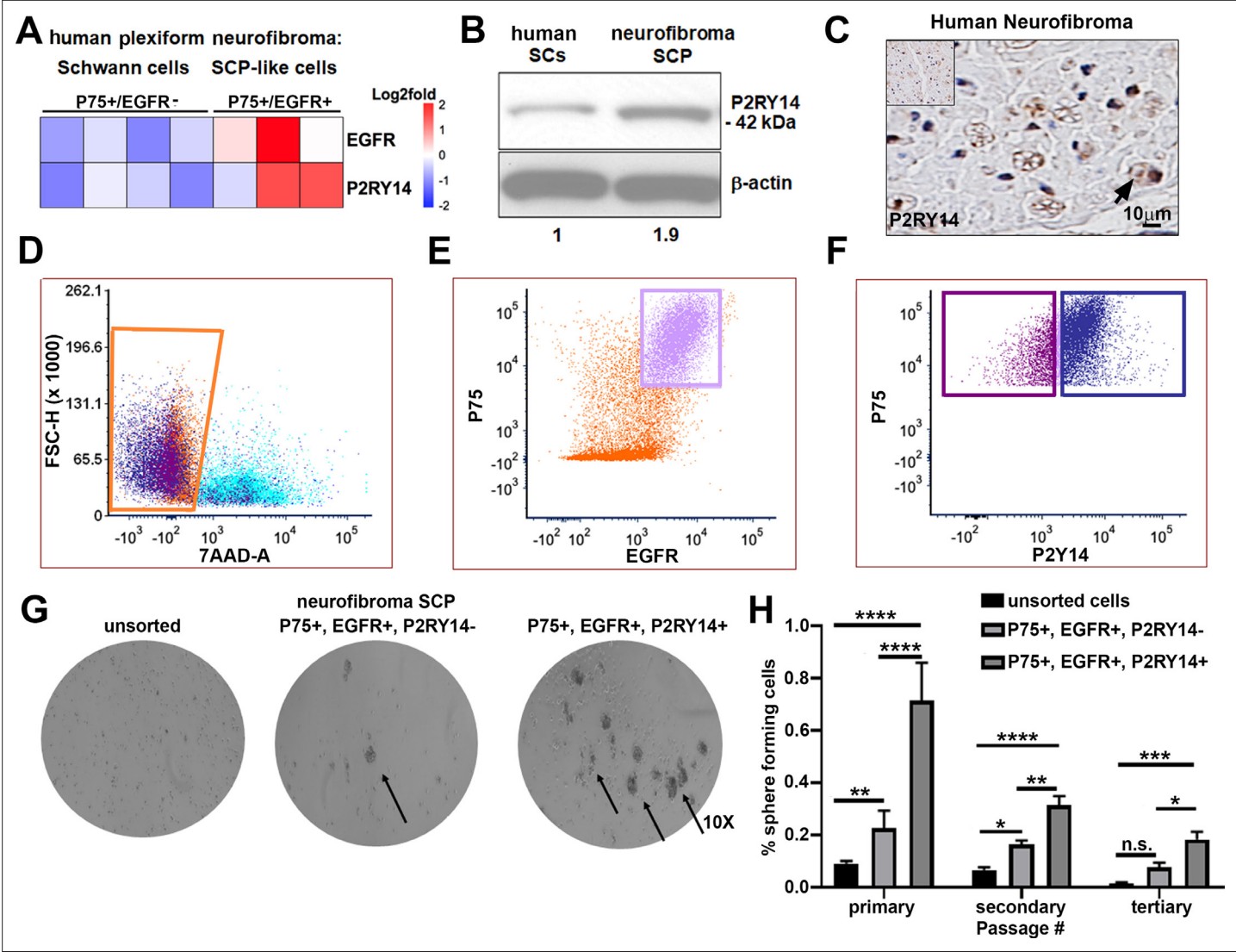

**Figure 1.** P2RY14 is expressed in human neurofibromas and promotes Schwann cell precursor (SCP) self-renewal in vitro. (**A**) Microarray heatmap shows *P2ry14* receptor expression in p75+/EGFR+ SCP-like tumor-initiating cells derived from human plexiform neurofibroma tumor cells compared to p75+/EGFR- SCP-like cells. (**B**) Western blot of human Schwann cells and neurofibroma SCP shows the latter has a 1.9-fold increase in *P2ry14* protein expression. (**C**) Immunohistochemistry of human neurofibroma shows *P2ry14* expression (DAB staining: brown [*P2ry14* positive cells] blue [cell nuclei]). (**D**) Representative fluorescence-activated cells sorting (FACS) plot shows live sorted human plexiform neurofibroma tumor cells. (**E**) Representative FACS plot shows human plexiform neurofibroma tumor cells sorted into p75+/EGFR+ SCP-like tumor-initiating cells (pink square). (**F**) Representative FACS plot shows p75+/EGFR+ SCP-like tumor-initiating cells further sorted into p75+/EGFR+/*P2ry14*- (left, purple square) and P75+/EGFR+/*P2ry14*+ (right, blue square). (**G**) Photomicrographs of human neurofibromas dissociated using FACS to yield: unsorted, p75+/EGFR+/*P2ry14*- and P75+/EGFR+/*P2ry14*+ cells. (**H**) Quantification of unsorted, p75+/EGFR+/*P2ry14*- and P75+/EGFR+/*P2ry14*+ cells plated in sphere medium. (n = 3; two-way ANOVA; primary: **p = 0.0057, ****p < 0.0001; secondary: *p = 0.0487; **p < 0.0024, ****p < 0.0001; tertiary: *p = 0.0321, ***p = 0.0006).

*NF1* deficient SCP self-renewal in vitro and reduced SC proliferation in neurofibroma bearing mice in vivo. We suggest that targeting the *P2ry14* receptor pathway could be relevant for treatment of NF1.

## Results

### P2RY14 is expressed in human neurofibroma SCP and promotes SCP self-renewal in vitro

To characterize SCPs we used flow cytometry. We dissociated cells from human PNs resected for therapeutic purposes from three neurofibroma patients. Cells were sorted into p75+/EGFR- SCs and

p75[+]/EGFR[+] SCP-like tumor-initiating cells. We performed gene expression analysis on these cells and found that *P2ry14* mRNA is elevated in p75[+]/EGFR[+] SCP-like tumor-initiating cells (*Figure 1A*). Western blot also showed expression in human SCs and neurofibroma SCPs, with 1.9-fold increase of P2ry14 protein in SCP-like cells (*Figure 1B*). Anti-*P2ry14* staining also showed *P2ry14* protein membranes of myelinating SCs in human neurofibroma tissue sections (*Figure 1C*). To test if *P2ry14*[+] SCP-like cells derived from human neurofibromas have altered ability to self-renew, we performed fluorescence-activated cells sorting (FACS) and sorted SCP-like cells into p75[+]/EGFR[+]/*P2ry14*[-] and p75[+]/EGFR[+]/*P2ry14*[+] cells and plated them at low density to generate unattached spheres in vitro (*Figure 1D–F*). Unsorted neurofibroma cells rarely form SCP-like spheres. FACS analysis (of cells from three additional neurofibroma tumors) showed that on average p75[+]/EGFR[+]/*P2ry14*[-] cells formed spheres at a frequency of 23.4%, while 64.8% of p75[+]/EGFR[+]/*P2ry14*[+] cells formed spheres. The p75[+]/EGFR[+]/*P2ry14* [+] cells maintained their significantly enhanced ability to form spheres in vitro for three passages (*Figure 1G and H*). Thus, *P2ry14* is overexpressed in human neurofibroma SCPs in vitro, and marks SCP with the potential to self-renew in vitro.

## Mouse *Nf1* mutant SCPs P2RY14-cAMP signaling regulates self-renewal

To test if *P2ry14* signaling is relevant in cultured SCP spheres from wild-type (WT) and *Nf1*[-/-] mouse embryos, we treated these cells with the highly selective *P2ry14* inhibitor PPTN (4-[4-(4-piperidinyl) phenyl]-7-[4-(trifluoromethyl)phenyl]-2-naphthalenecarboxylic acid) hydrochloride (*Robichaud et al., 2011*). We found that treatment with PPTN reduced the percentage of *Nf1*[-/-] mouse SCPs that formed spheres, but had little effect on WT SCPs, consistent with a role in *Nf1* mutant SCP self-renewal (*Figure 2A*). Photographs of spheres are shown in *Figure 2—figure supplement 1A*; this experiment was repeated in three biological replicates with similar results. Dose-response analysis confirmed that the optimal concentration of the *P2ry14* inhibitor (PPTN) is 300 nM; as 500 nM PPTN was toxic (*Figure 2—figure supplement 2A–F*).

To confirm these results, we silenced *P2ry14* gene expression using short-hairpin RNAs (shRNAs) targeting *P2ry14*. WT and *Nf1*[-/-] mouse SCPs were treated with non-target control (shNT) or shRNA *P2ry14* (sh*P2ry14*). sh*P2ry14* treated cells showed reduced *P2ry14* mRNA (*Figure 2B*) and P2ry14 protein (*Figure 2C*). Sphere formation was significantly decreased in *Nf1*[-/-] SCPs but not WT treated with *P2ry14* shRNA, and this phenotype also persisted for three passages (*Figure 2D*). Photomicrographs are shown for one shRNA (sh09) (*Figure 2—figure supplement 1B*); but the experiment was repeated with two additional *P2ry14* shRNAs in three biological replicates each, with similar results (*Figure 2—figure supplement 3A-D*). We also analyzed cAMP-dependent protein kinase (PKA) substrate phosphorylation (using anti-p-PKA substrate antibody) as an indirect read out of cAMP levels in cells. p-PKA substrate phosphorylation increased in WT and *Nf1*[-/-] SCPs sh*P2ry14* treated cells. Importantly, in *Nf1*[-/-] SCPs knockdown of *P2ry14* increased levels of pPKA to those in WT shNT levels (*Figure 2E*). Thus, PPTN or shRNA targeting P2ry14 affects *Nf1* mutant SCP spheres more than WT spheres, a differential effect consistent with deregulated cell signaling in Nf1 mutant cells.

To understand if *P2ry14*-mediated changes in cAMP signaling affect SCP self-renewal, we treated WT and *Nf1*[-/-] SCPs with the specific phosphodiesterase-4 (PDE4) inhibitor, rolipram. Rolipram blocks degradation of cAMP by PDE4, increasing intracellular levels of cAMP (*Mackenzie and Houslay, 2000*). Treatment with 1 µM rolipram or 300 nM of the *P2ry14* inhibitor (PPTN) decreased *Nf1*[-/-] SCP self-renewal; the combination showed additional effect, largely at early passage (*Figure 2F*). Photographs of these experiments are shown in (*Figure 2—figure supplement 1C*). These results support the idea that the elevated self-renewal in *Nf1*[-/-] SCP is due, at least in part, to P2ry14 G$_i$-mediated changes in cAMP.

## P2RY14 is expressed in vivo in mouse SCPs and SCs

In the *Nf1*[fl/fl];*Dhh*[Cre] neurofibroma mouse model, Cre recombinase is expressed from Desert Hedgehog (Dhh) regulatory sequences to effect recombination of the *Nf1*[fl/fl] allele, resulting in loss of both *Nf1* alleles in developing SCPs at E12.5 (*Wu et al., 2008*). *Nf1*[fl/fl];*Dhh*[Cre] mice develop paraspinal neurofibromas that have loss of axon-SC interaction in Remak bundles, mast cell and macrophage accumulation, and nerve fibrosis, all characteristics of human PNs (*Wu et al., 2008*; *Liao et al., 2018*; *Fletcher et al., 2019a*; *Fletcher et al., 2019b*). In this setting, PNs are present by 4 months of age,

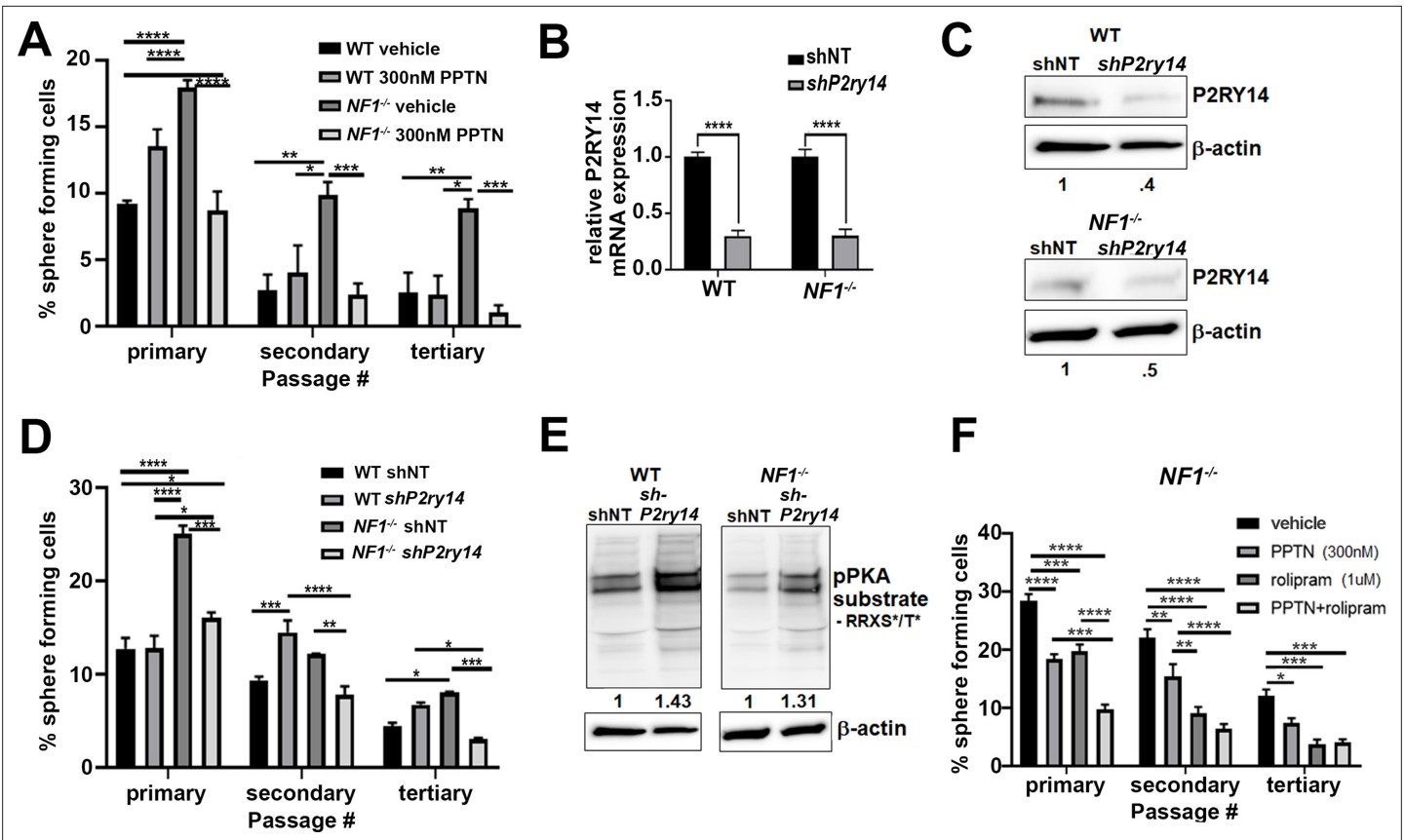

**Figure 2.** Mouse *Nf1* mutant Schwann cell precursors (SCPs) use P2RY14 signaling to regulate self-renewal. (**A**) Quantification of percent of sphere forming cells in mouse wild-type (WT) and *Nf1⁻/⁻* SCPs treated with the selective *P2ry14* inhibitor (300 nM 4-[4-(4-piperidinyl)phenyl]-7-[4-(trifluoromethyl) phenyl]-2-naphthalenecarboxylic acid [PPTN]) (primary, secondary, tertiary passage) (n = 3; two-way ANOVA; primary: *p = 0.0375, ***p = 0.0001, ****p < 0.0001; secondary: *p = 0.0101, **p = 0.0015, ***p = 0.0009; tertiary: *p = 0.0101, **p = 0.0050, ***p = 0.0005). (**B**) *P2ry14* mRNA expression in WT and *Nf1⁻/⁻* E12.5 mouse SCP treated with sh non-target (shNT) control and sh*P2ry14* (****p < 0.0001). (**C**) Western blot of WT and *Nf1⁻/⁻* SCPs treated with shNT and sh*P2ry14* showing *P2ry14* knockdown. WT sh*P2ry14* show a 0.4-fold decrease of *P2ry14* protein compared to WT shNT. *Nf1⁻/⁻* sh*P2ry14* show a 0.5-fold decrease compared to *Nf1⁻/⁻*. (**D**) Quantification of percent of sphere forming cells in mouse WT and *Nf1⁻/⁻* SCPs treated with shNT and sh*P2ry14* (n = 3; two-way ANOVA; primary: *p = 0.0288, ****p < 0.0001; secondary: **p = 0.0029,***p = 0.0005, ****p < 0.0001; tertiary: *p = 0.0154, ***p = 0.0007). (**E**) Western blot of WT and *Nf1⁻/⁻* Schwann cell (SC) spheres shows changes in pPKA substrate phosphorylation after sh*P2ry14* knockdown. WT sh*P2ry14* shows a 1.43-fold increase in pPKA after *P2ry14* knockdown. *Nf1-/-* cells have a 1.31-fold increase in pPKA expression after *P2ry14* knockdown. (**F**) Quantification of percent of sphere forming cells in *Nf1⁻/⁻* mouse SCPs treated with 1 µM rolipram or 300 nM PPTN (n = 3; two-way ANOVA; primary: ***p = 0.0002, ****p < 0.0001; secondary **p = 0.0030, ****p < 0.0001; tertiary: *p = 0.0476, ***p = 0.0004).

The online version of this article includes the following figure supplement(s) for figure 2:

**Figure supplement 1.** Photomicrographs of three different experiments in which wild-type (WT) and *Nf1⁻/⁻* Schwann cell precursors (SCPs) were treated with 4-[4-(4-piperidinyl)phenyl]-7-[4-(trifluoromethyl)phenyl]-2-naphthalenecarboxylic acid (PPTN), sh*P2ry14* and rolipram.

**Figure supplement 2.** 4-[4-(4-Piperidinyl)phenyl]-7-[4-(trifluoromethyl)phenyl]-2-naphthalenecarboxylic acid (PPTN) dose-response analysis in wild-type (WT) and *Nf1-/-* Schwann cell precursors (SCPs).

**Figure supplement 3.** Photomicrographs and quantification analysis of two biological replicates of sh*P2ry14* in wild-type (WT) and *Nf1⁻/⁻* Schwann cell precursors (SCPs).

and neurofibromas enlarge as the mice age, ultimately compressing the spinal cord causing paralysis and thus, necessitating sacrifice (***Wu et al., 2008***).

We bred *P2ry14⁻/⁻* mice (***Meister et al., 2014***) and generated *P2ry14⁻/⁻; Nf1ᶠˡ/ᶠˡ;Dhhᶜʳᵉ, P2ry14⁺/⁻; Nf1ᶠˡ/ᶠˡ;Dhhᶜʳᵉ* littermates, and *Nf1ᶠˡ/ᶠˡ;Dhhᶜʳᵉ* controls (***Figure 3A***). Genotyping and western blotting verified reduced P2ry14 in *P2ry14⁻/⁻; Nf1ᶠˡ/ᶠˡ;Dhhᶜʳᵉ* sciatic nerve and neurofibroma tumors compared to *Nf1ᶠˡ/ᶠˡ;Dhhᶜʳᵉ* controls (***Figure 3B and C***). In the *P2ry14⁻/⁻* mice, most of the coding region of *P2RY14* is replaced by a β-galactosidase cassette, so that where the *P2ry14* gene is expressed, β-galactosidase is detectable (***Meister et al., 2014***). We visualized *P2ry14* in SOX-10 positive SCPs in the spinal cord

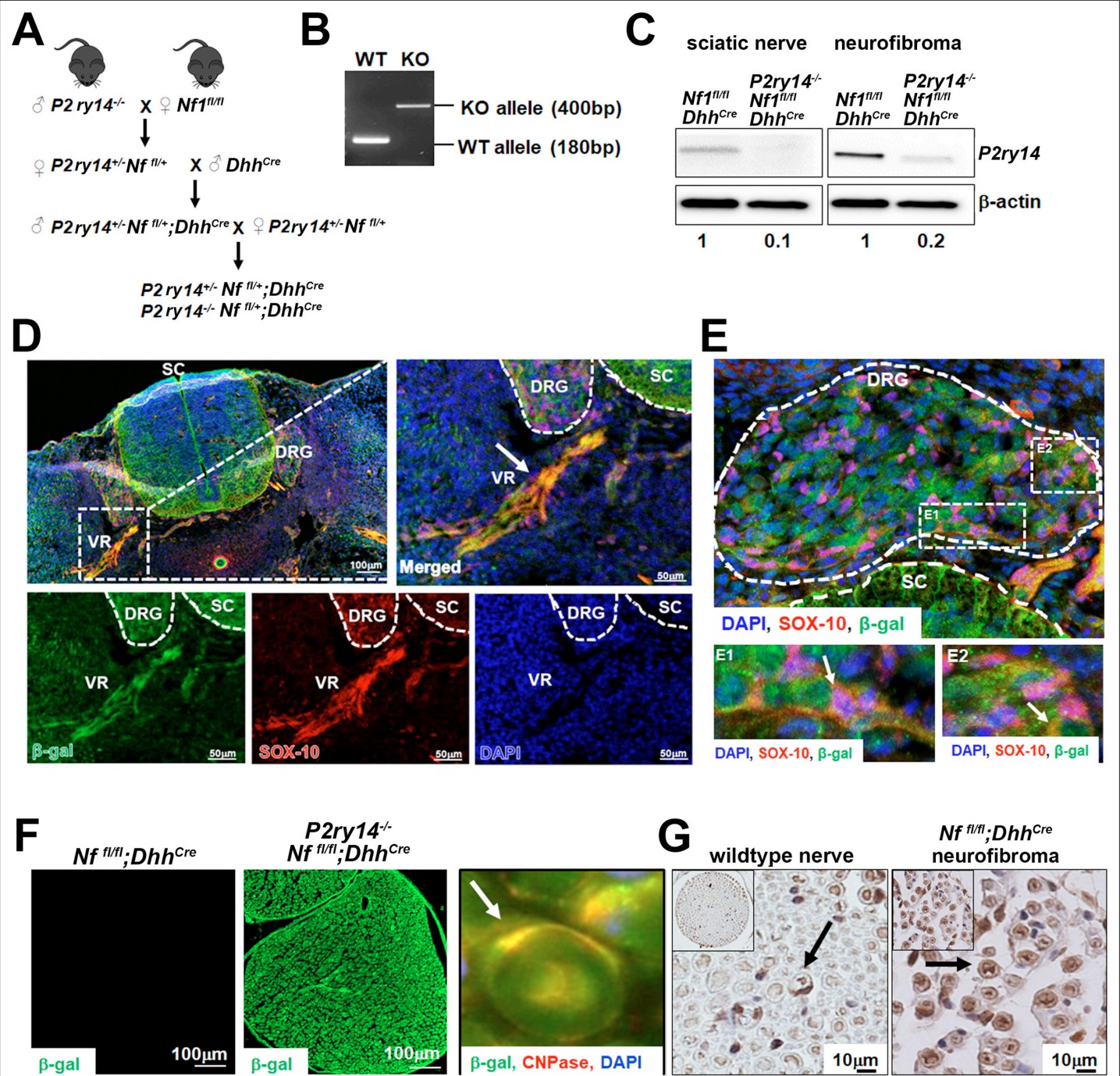

**Figure 3.** P2RY14 is expressed in Schwann cell precursors (SCPs) and Schwann cells (SCs) in vivo in a mouse model of neurofibromatosis. (**A**) Schematic of generation of neurofibroma mice breeding of *P2ry14^-/-* mice with *Nf^fl/fl* mice to obtain *P2ry14^+/-; Nf1^fl/fl;Dhh^Cre* and *P2ry14^-/-; Nf1^fl/fl;Dhh^Cre* littermates after several crosses. (**B**) Genotyping confirmation of wild-type (WT) and *P2ry14* knockout (KO) alleles. (**C**) Western blot of sciatic nerve and neurofibroma tumors of *Nf1^fl/fl;Dhh^Cre* and *P2ry14^-/-;Nf1^fl/fl;Dhh^Cre* mice show decrease in *P2ry14* expression upon *P2ry14* knockdown (1- to 0.1-fold decrease in sciatic nerve and 1- to 0.2-fold decrease in neurofibroma tissue). (**D**) Spinal cord (SC) immunofluorescent staining of mouse embryos at embryonic day 12.5 (E12.5) shows *P2ry14* expression (β-galactosidase) co-localization with SOX-10 SCs at the dorsal root ganglion (DRG) and ventral roots (VR). (**E**) Spinal cord (SC) immunofluorescent staining of mouse embryos at E12.5 shows *P2ry14* expression (β-galactosidase) co-localization with SOX-10 SCs at DRG. E1 and E2 insets show enlarged sections of the DRG. (**F**) Immunofluorescent staining of 7-month-old mouse sciatic nerve shows β-galactosidase positive staining as a confirmation of *P2ry14* knock-in; co-labeling of β-galactosidase and CNPase shows that *P2ry14* co-localizes with SCs (inset). (**G**) DAB staining of 7-month-old WT nerve and *Nf1^fl/fl;Dhh^Cre* mouse neurofibromas (DAB staining: brown [*P2ry14* positive cells] blue [cell nuclei]).

dorsal and ventral roots (VR) and in the dorsal root ganglion (DRG) of E12.5 mice (*Figure 3D and E*). *P2ry14⁻/⁻; Nf1ᶠˡ/ᶠˡ;Dhhᶜʳᵉ* SCs in 7-month-old mice were also positive for β-galactosidase staining (*Figure 3F*). CNPase (2′,3′ cyclic nucleotide 3′ phosphodiesterase), a known SC marker, co-localized with β-galactosidase staining in the sciatic nerve (*Figure 3F*, inset). *P2ry14* antibody staining confirmed *P2ry14* expression in myelinating SC in mouse WT nerve and in *Nf1ᶠˡ/ᶠˡ;Dhhᶜʳᵉ* neurofibromas (*Figure 3G*).

## P2RY14 deletion in mouse model of neurofibroma increases survival, delays neurofibroma initiation, and improves SC Remak bundle disruption

Kaplan-Meier survival analysis showed that *P2ry14⁻/⁻; Nf1ᶠˡ/ᶠˡ;Dhhᶜʳᵉ* have a significant survival advantage compared to neurofibroma-bearing mice (*Nf1ᶠˡ/ᶠˡ;Dhhᶜʳᵉ*; p = 0.0256). In fact, *P2ry14⁻/⁻;Nf1ᶠˡ/ᶠˡ;Dhhᶜʳᵉ* did not differ significantly from non-neurofibroma bearing *Nf1ᶠˡ/ᶠˡ;Dhhᶜʳᵉ* control littermates (p = 0.1367) (*Figure 4A*). To test if *P2RY14* deletion plays a role in neurofibroma initiation, we analyzed *Nf1ᶠˡ/ᶠˡ;Dhhᶜʳᵉ* and *P2ry14 ⁻/⁻;Nf1ᶠˡ/ᶠˡ;Dhhᶜʳᵉ* mice to 4 months of age, when tumor is first detectable. Gross dissection of spinal cords from these mice showed that *P2ry14⁻/⁻;Nf1ᶠˡ/ᶠˡ;Dhhᶜʳᵉ* have decreased neurofibroma number, consistent with a role in tumor initiation, and only slightly reduced neurofibroma diameter (*Figure 4—figure supplement 1A–C*). Ki67 and H&E staining of neurofibromas in these 4-month-old mice showed no evident changes between genotypes in cell proliferation or cell morphology (*Figure 4—figure supplement 1D–F*).

To determine if the effects of *P2ry14* differ over time, we aged *Nf1ᶠˡ/ᶠˡ;Dhhᶜʳᵉ* and *P2ry14⁻/⁻;Nf1ᶠˡ/ᶠˡ;Dhhᶜʳᵉ* mice. At 7 months *P2ry14⁻/⁻;Nf1ᶠˡ/ᶠˡ;Dhhᶜʳᵉ* mice also showed significantly fewer neurofibromas compressing the spinal cord on gross dissection (*Figure 4B and C*); tumor diameter was reduced to a lesser degree (*Figure 4D*). At this age, *P2ry14* loss decreased Ki67+ cells in neurofibroma tissue sections (*Figure 4E–F*); many of these proliferative cells expressed the SC marker CNPase (*Figure 4—figure supplement 1G*). However, H&E staining showed characteristic neurofibroma morphology (*Figure 4—figure supplement 1H*). To determine if cAMP is affected by loss of *P2ry14* in tumors, we stained with anti-p-PKA substrate antibody. *Nf1ᶠˡ/ᶠˡ;Dhhᶜʳᵉ* nerves showed a decrease in p-PKA substrate phosphorylation versus WT mice. Remarkably, this decrease was reversed in *P2ry14⁻/⁻;Nf1ᶠˡ/ᶠˡ;Dhhᶜʳᵉ* nerves (*Figure 4G*). pPKA labeling co-localized with CNPase, suggesting that the changes in p-PKA substrate phosphorylation expression are at least in part, in nerve SCs (*Figure 4G*, insets). Based on these results, we conclude that *P2ry14* deletion in vivo in neurofibroma mice increases mouse survival and delays neurofibroma initiation and has lesser effects on SC proliferation.

We also examined the effects of *P2ry14* loss on nerve disruption phenotype using electron microscopy. At 4 months, *Nf1ᶠˡ/ᶠˡ;Dhhᶜʳᵉ* nerves already show disrupted Remak bundles (reduced numbers of axons ensheathed by individual SCs); this phenotype was rescued in *P2ry14⁻/⁻;Nf1ᶠˡ/ᶠˡ;Dhhᶜʳᵉ* mice (*Figure 5A and C*). By 7 months, *Nf1ᶠˡ/ᶠˡ;Dhhᶜʳᵉ* nerves were even more severely disrupted, and rescued to near WT levels by *P2ry14* loss (*Figure 5B and D*). We conclude that genetic knockout of *P2ry14* in neurofibroma-bearing mice improves the Remak bundle defects characteristic of neurofibroma-bearing mice, and that are present in both nerve and neurofibroma.

## Increasing cAMP in neurofibroma-bearing mice by rolipram or a P2RY14 inhibitor decreases SC proliferation in neurofibromas

Results presented above indicate that loss of *P2ry14* during embryonic development delays phenotypes caused by *Nf1* loss in SCP and SCs. To test if *P2ry14* signaling might also play roles in tumor maintenance, we took pharmacological approaches. We treated *Nf1ᶠˡ/ᶠˡ;Dhhᶜʳᵉ* mice with vehicle or 5 mg/kg rolipram for 60 days to test if cell proliferation in established *Nf1ᶠˡ/ᶠˡ;Dhhᶜʳᵉ* neurofibromas is affected by increasing cAMP levels (*Figure 6A*). All mice survived rolipram treatment without significant weight loss. Neurofibroma lysates from rolipram treated mice confirmed increases in p-PKA substrate phosphorylation (*Figure 6B*). Numbers of mice assessed did not allow for tumor volume analysis. However, cell proliferation (Ki67 staining) in tissue sections revealed that rolipram treated neurofibromas show significantly decreased cell proliferation in vivo (*Figure 6C and D*).

To test if short-term inhibition of *P2ry14* in *Nf1ᶠˡ/ᶠˡ;Dhhᶜʳᵉ* neurofibroma mice similarly affects pPKA levels and cell proliferation, we treated 4-month-old *Nf1ᶠˡ/ᶠˡ;Dhhᶜʳᵉ* mice with a *P2ry14* inhibitor (PPTN) (*Robichaud et al., 2011*; *Battistone et al., 2020*). Oral bioavailability for this inhibitor is poor,

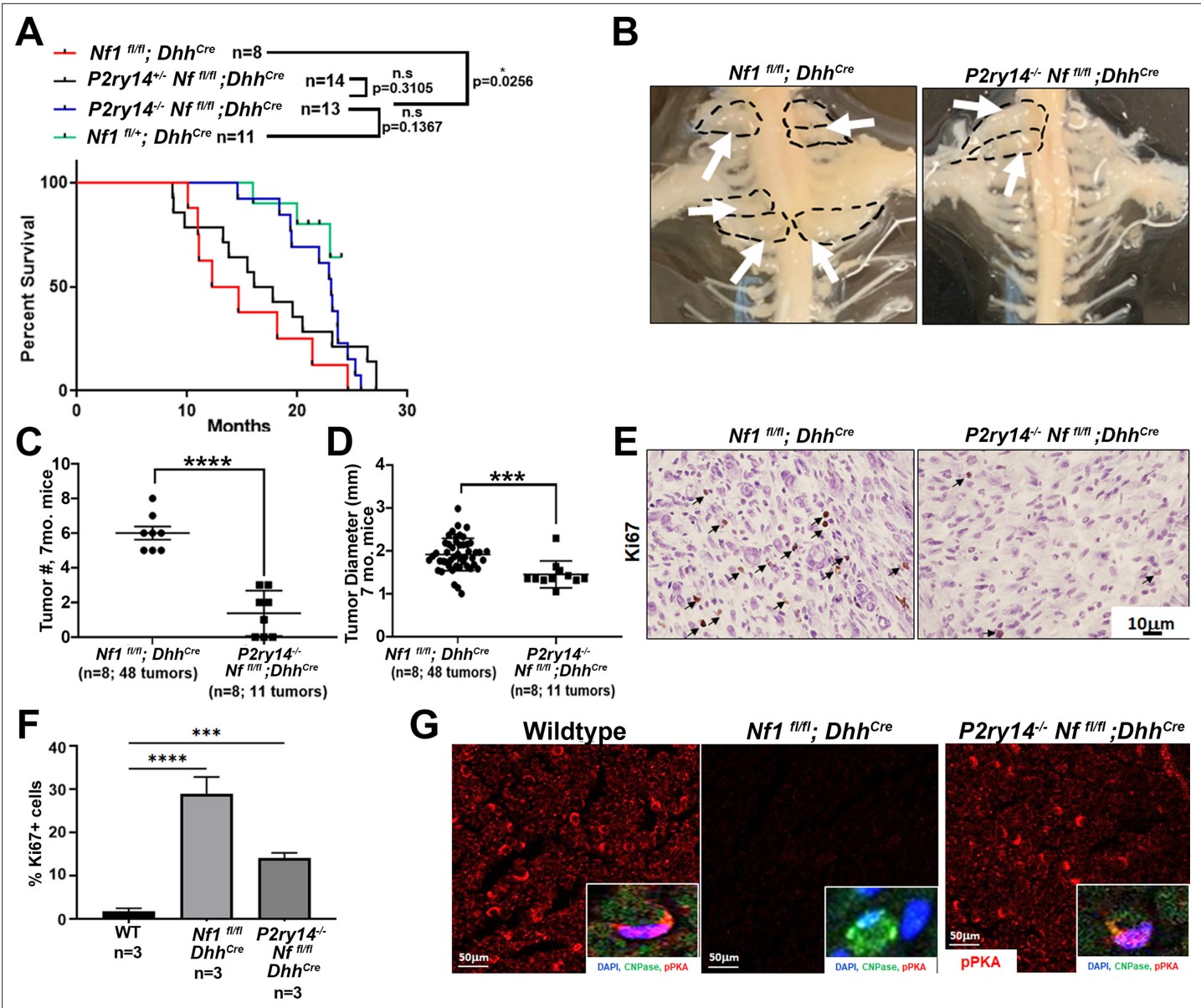

**Figure 4.** P2RY14 deletion in a mouse model of neurofibroma increases survival and delays neurofibroma initiation. (**A**) Kaplan-Meier survival plot of *Nf1fl/fl;DhhCre* (red line; n = 8); *P2ry14+/-; Nf1fl/fl;DhhCre* (black line, n = 14); *P2ry14-/-; Nf1fl/fl;DhhCre* (blue line, n = 13); *Nf1fl/+* control (green line, n = 11) (*p = 0.0256) shows *P2ry14-/-;Nf1fl/fl;DhhCre* have increased survival. (**B**) Representative image of gross dissection of *Nf1fl/fl;DhhCre* and *P2ry14-/-;Nf1fl/fl;DhhCre* mice at 7 months of age. (**C**) Neurofibroma tumor number quantification at 7 months of age (unpaired t-test ****p < 0.0001). (**D**) Neurofibroma diameter quantification at 7 months (unpaired t-test ***p = 0.0004) (for **C** and **D**: *Nf1fl/fl;Dhh+* n = 8 mice, 48 neurofibroma tumors; *P2ry14-/-;Nf1fl/fl;Dhh+* n = 8 mice, 11 neurofibroma tumors). (**E**) Ki67 staining of mouse dorsal root ganglion (DRG) and neurofibroma tumor tissue at 7 months of age. (**F**) Quantification of Ki67+ cells in mouse DRG and neurofibroma tumor tissue at 7 months of age (one-way ANOVA; multiple comparisons ***p = 0.0008; ****p < 0.0001). (**G**) Anti-p-PKA substrate staining in wild-type (WT), *Nf1fl/fl;DhhCre* and *P2ry14-/-; Nf1fl/fl;DhhCre* mice. p-PKA substrate phosphorylation labeling co-localized with CNPase Schwann cell (SC) marker (insets).

The online version of this article includes the following figure supplement(s) for figure 4:

**Figure supplement 1.** Tumor dissection of Nf1fl/fl;DhhCre and P2ry14-/-;Nf1fl/fl;DhhCre mice at 4 months and immunostaining analysis of 4- and 7-month sciatic nerve and tumors.

therefore, osmotic minipumps were implanted subcutaneosuly to release a daily dose of 4.55 mg/kg for 14 days as described (***Battistone et al., 2020***). At day 14, mice were sacrificed and tissue harvested (***Figure 6E***). Immunostaining showed the expected increase in pPKA-substrate staining in the sciatic nerve of *Nf1fl/fl;DhhCre* mice after PPTN treatment (***Figure 6F***). Short-term treatment with

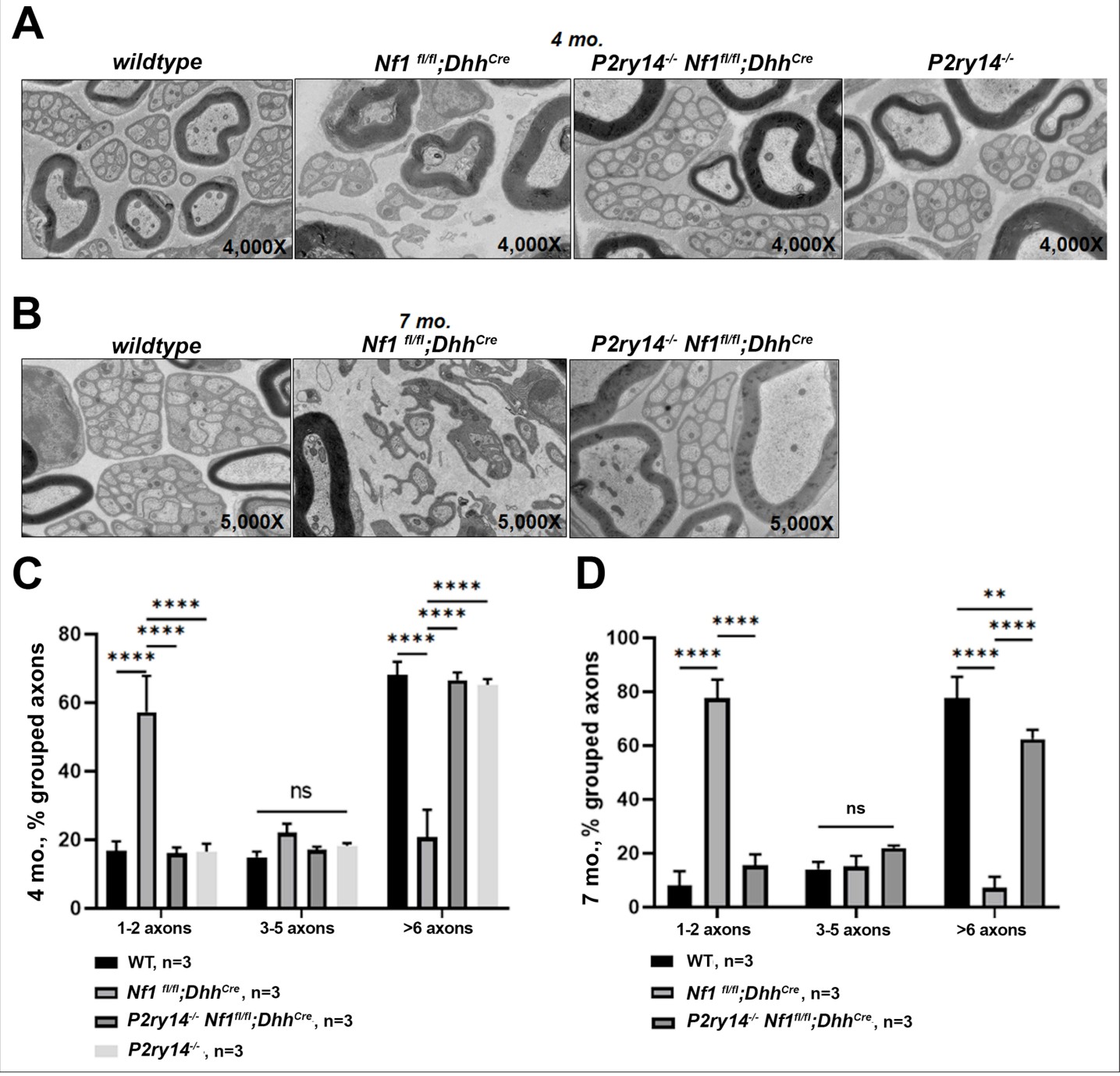

**Figure 5.** P2RY14 deletion improves nerve ultrastructure. (**A**) Electron micrograph of saphenous nerve of 4-month-old wild-type (WT), *Nf1*$^{fl/fl}$*;Dhh*$^{Cre}$, *P2ry14*$^{-/-}$; *Nf1*$^{fl/fl}$*;Dhh*$^{Cre}$ and *P2ry14*$^{-/-}$ mice. (**B**) Electron micrograph of saphenous nerve of 7-month-old WT, *Nf1*$^{fl/fl}$*;Dhh*$^{Cre}$ and *P2ry14*$^{-/-}$*;Nf1*$^{fl/fl}$*;Dhh*$^{Cre}$ mice. (**C**) Remak bundle quantification at 4 months of age (n = 3; two-way ANOVA: ****$p < 0.0001$). (**D**) Remak bundle quantification at 7 months of age (n = 3; two-way ANOVA: **$p = 0.0027$, ****$p < 0.0001$).

the *P2ry14* inhibitor also decreased cell proliferation (Ki67+ cells) in neurofibroma tissue sections PPTN treated mice (*Figure 6G and H*). These results suggest that *P2ry14* and cAMP play at least a partial role in tumor maintenance. Altogether, these in vitro and in vivo studies show that in a mouse model of neurofibromatosis, *P2ry14* is a key regulator of SCP self-renewal, SC proliferation, neurofibroma initiation, and neurofibroma maintenance.

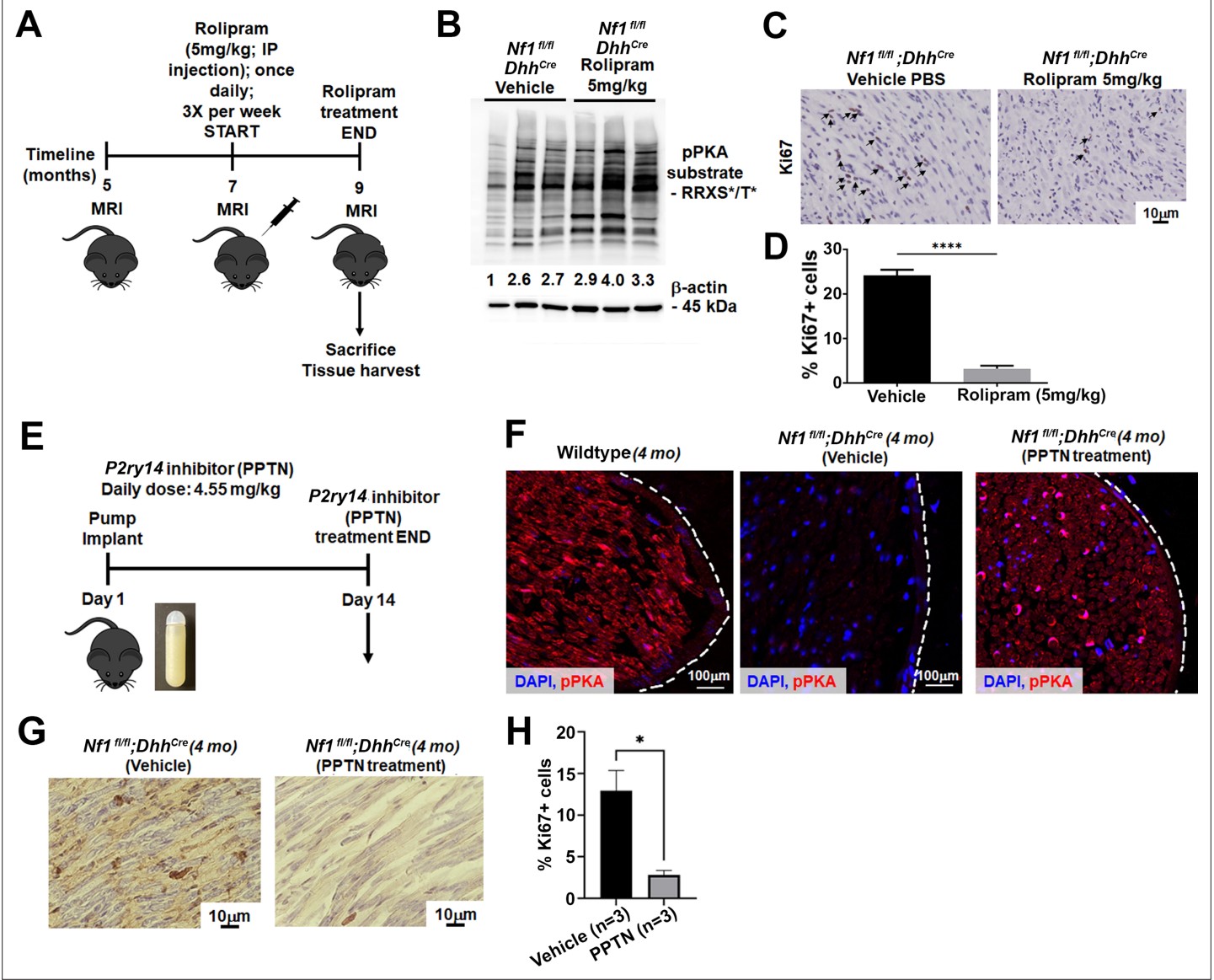

**Figure 6.** Cyclic AMP (cAMP) increase in a mouse model of neurofibromatosis either by rolipram or P2RY14 inhibitor treatment decreases Schwann cell (SC) proliferation. (**A**) Rolipram drug treatment experimental design. (**B**) Tumor lysates of vehicle and rolipram treated *Nf1^fl/fl^;Dhh^Cre^* mice show changes in p-PKA substrate. (**C**) Ki67 staining at 9 months of age in vehicle and rolipram treated mice. (**D**) Quantification of Ki67+ cells in vehicle treated versus rolipram treated mice (unpaired t-test: ****p < 0.0001; n = 3). (**E**) *P2ry14* inhibitor (4-[4-(4-piperidinyl)phenyl]-7-[4-(trifluoromethyl)phenyl]-2-naphthalenecarboxylic acid [PPTN]) drug treatment experimental design. (**F**) Immunofluorescent staining of sciatic nerve of 4-month-old wild-type (WT), *Nf1^fl/fl^;Dhh^Cre^* (vehicle) and *Nf1^fl/fl^;Dhh^Cre^* (PPTN treated) shows increased p-PKA expression after PPTN treatment. (**G**) Ki67+ staining of *Nf1^fl/fl^;Dhh^Cre^* (vehicle) and *Nf1^fl/fl^;Dhh^Cre^* (PPTN treated) neurofibroma tissue. (**H**) Quantification of Ki67+ cells after PPTN treatment in neurofibroma tissue.

## Discussion

GPCR-mediated regulation of cAMP occurs upon *NF1* loss in mammals (*Dasgupta et al., 2003*; *Deraredj Nadim et al., 2016*), fish, and dropsophila (*Tong et al., 2002*; *Wolman et al., 2014*). In these systems, PACAP receptors and serotonin receptors have been identified as GPCRs that act upstream of NF1 (*Anastasaki and Gutmann, 2014*; *Deraredj Nadim et al., 2016*). However, the potential role of cAMP in neurofibroma remains unclear. Targeting GPCR signaling has been suggested as a potential therapeutic option to treat NF1, so exploring the relevance of this pathway to peripheral nerve tumors is important.

We found that human neurofibroma-derived SCP-like cells sorted for GPCR *P2ry14* expression have increased self-renewal potential. Adult peripheral nerves do not contain a stem cell population

(*Stierli et al., 2018*). Therefore, neurofibroma SCP-like cells may result from persistence of immature cells and/or from de-differentiation of mutant SCs. In either case, neurofibroma also contain cells that also express *P2ry14*. Our findings are entirely consistent with findings that hematopoietic stem/progenitor cells marked by *P2ry14* stimulate self-renewal (*Lee et al., 2003*; *Cho et al., 2014*; *Holmfeldt et al., 2016*).

Purinergic receptor *P2ry14* is activated by UDP and by the nucleotide sugars UDP-glucose, UDP-galactose, UDP-glucuronic acid, and UDP-*N*-acetylglucosamine (*Chambers et al., 2000*; *Moore et al., 2003*; *Abbracchio et al., 2003*; *Carter et al., 2009*; *Conroy et al., 2016*). *NF1* mutant cells may release one or more of these ligands, because SCP self-renewal in serum-free medium was reduced by pharmacological *P2ry14* inhibition, and by shRNA targeting *P2ry14*, in the absence of added UDP or UDP-sugars. The small increase in *P2ry14* protein in mutant cells may contribute to increasing signaling downstream of the GPCR, and/or *Nf1* deficient cells may release more UDP-sugars than WT cells. While it is difficult to measure UDP levels in the extracellular milieu without causing cell damage and concomitant release of UDP-sugars (Lazarowki & Harden, 2015), it will be of interest to measure UDP and UDP-sugars both in the neurofibroma extracellular milieu, and in SCP and SC culture medium.

UDP, UTP, and other nucleotide sugars including UDP-glucose are present at high levels in tumor cells, and released from cells in tumors and injury sites, where they act as danger signals that trigger inflammatory responses (*Eigenbrodt et al., 1992*; *Skelton et al., 2003*). We focused on the role of *P2ry14* in SCs and SCPs, because most of the *P2ry14* expression in peripheral nerve neurofibromas is in CNPase$^+$ myelinating SCs, based on our use of a β-galactosidase reporter and antibody staining. However, *P2ry14* is also expressed in immune cells and in small sensory neurons and larger diameter sensory neurons (*Skelton et al., 2003*; *Müller et al., 2005*; *Scrivens and Dickenson, 2005a*; *Scrivens and Dickenson, 2006*; *Malin and Molliver, 2010*). In our studies, because we used a global knockout, we cannot exclude the idea that some of the observed in vivo effects are SCP-independent. The recent description of a conditional knockout of *P2ry14* should enable additional studies (*Battistone et al., 2020*).

Whether the decreased levels of cAMP in *NF1* deficient cells requires upstream RAS signaling is unclear (reviewed in *Bergoug et al., 2020*). In *Nf1* mutant neurons, RAS activation of atypical protein kinase C-zeta causes GRK2-driven Gαs/AC inactivation downstream of the PACAP receptor (*Anastasaki and Gutmann, 2014*). In fly, some investigators proposed direct regulation of AC by loss of *Nf1* (*Tong et al., 2002*), but others propose that regulation is indirect (*Walker and Bernards, 2014*). In addition, PKA phosphorylates neurofibromin and inhibits its GAP activity (*Feng et al., 2004*), suggesting potential negative feedback between NF1/RAS and PKA in WT cells. Furthermore, in some cell types, downstream of *P2ry14*/Gi/o, Ca2$^+$ mobilization can result in release of factors such as IL-18, which could activate RAS indirectly (*Moore et al., 2003*; *Arase et al., 2009*). All these mechanisms are candidates to explore for relevance to *P2ry14* signaling in SCPs and SCs.

Genetic knockdown of *P2ry14* increased levels of cAMP-dependent protein kinase (PKA)-mediated phosphorylation in SCP in vitro and SC in vivo, and *NF1*$^{-/-}$ SCPs treated with rolipram to increase cAMP showed decreased SCP self-renewal. In vivo, neurofibroma bearing mice treated with either rolipram or the *P2ry14* inhibitor showed increases in pPKA expression and decreases in SC proliferation after treatment. These results implicate cAMP modulation as a critical *P2ry14* effector in SCs and SCPs.

Consistent with these findings, in WT cells, *P2ry14*/Gi/o-mediated decreases in cAMP affect chemotaxis hematopoietic of stem/progenitor cells (*Lee et al., 2003*; *Scrivens and Dickenson, 2005b*). Supporting a role for *P2ry14*-G$_i$-driven suppression of cAMP levels in SCP/SC, p-PKA substrate staining was reduced in *Nf1* mutant nerves and elevated by additional loss of *P2ry14*. Importantly, increasing cellular cAMP levels with rolipram mimicked the effects of inhibiting *P2ry14*, reducing SCP self-renewal in vitro and SC proliferation in vivo. Our findings in peripheral nerve neurofibromas are entirely consistent with those in NF1 brain tumors. *Warrington et al., 2010*, found that overexpression of *PDE4A1* caused formation of hypercellular lesions with features of NF1-associated glioma, and rolipram inhibited optic glioma growth in an NF1-driven mouse model (*Warrington et al., 2010*). Thus, rolipram treatment may be relevant to multiple NF1 manifestations.

In conclusion, *P2ry14* cAMP signaling regulates SCP self-renewal and nerve tumor initiation in neurofibromatosis. *P2ry14* is a candidate for therapeutic intervention as a whole-body knockout of this receptor showed no effect on organismal survival or growth (*Xu et al., 2012*; *Meister et al.,*

*2014*). While the *P2ry14* inhibitor (PPTN) has poor PK properties (*Robichaud et al., 2011*), when drug candidates targeting *P2ry14* become available, they may be useful for prevention or treatment of NF1-driven neurofibromas.

## Materials and methods
### Resource availability
### Lead contact
Further information and requests for resources and reagents should be directed to and will be fulfilled by the Lead Contact, Nancy Ratner, PhD (nancy.ratner@cchmc.org).

### Materials availability
This study did not generate new unique reagents.

### Animals
We housed mice in a temperature and humidity-controlled vivarium on a 12 hr dark-light cycle with free access to food and water. The animal care and use committee of Cincinnati Children's Hospital Medical Center approved all animal use. WT C57Bl/6 mice were from Jackson Laboratory and were used at 4 and 7 months of age. The *Nf1*$^{fl/fl}$;DhhCre mouse line has been described previously by *Wu et al., 2008*. We bred *P2ry14*$^{-/-}$ male mice (*Meister et al., 2014*; Takeda Cambridge Limited) with *Nf1*$^{fl/fl}$ female mice to obtain the F1 generation *P2ry14*$^{+/-}$;*Nf1*$^{fl/+}$; then, we bred the F1 mice with *Nf1*$^{fl/fl}$;*Dhh*$^{Cre}$ male mice to obtain *P2ry14*$^{+/-}$;*Nf1*$^{fl/+}$;*Dhh*$^{Cre}$. Then, we bred *P2ry14*$^{+/-}$;*Nf1*$^{fl/+}$;*Dhh*$^{Cre}$ (males) with *P2ry14*$^{+/-}$;*Nf1*$^{fl/+}$; *Dhh*$^{Cre}$ (females) to obtain *P2ry14*$^{+/-}$; *Nf1*$^{fl/fl}$;*Dhh*$^{Cre}$ and *P2ry14;Nf1*$^{fl/fl}$;*Dhh*$^{Cre}$. We genotyped mice as described by *Meister et al., 2014*. Littermates were used for controls. Mice of both sexes were used for all experiments.

### Human neurofibroma sample collection
Fresh PNs (n = 6) were obtained after medically mandated surgeries. Mayo Clinic's IRB approved a Waiver for HIPAA Authorization and Informed Consent. The reasons given for a Waiver were (a) subject identifiers were removed, (b) research is minimal risk, (c) the waiver will not adversely affect the rights and welfare of the subjects, and (d) research could not be practicably carried out without the waiver.

### FACS analysis
Fresh surgical PN specimens were enzymatically dissociated as described (*Williams et al., 2008*). For cell sorting, we incubated cell suspensions with anti-*P2ry14* receptor antibody (Rabbit, polyclonal, Alomone labs, # APR-018; RRID:AB_2039847) on ice for 30 min, washed with PBS twice. We then incubated cells with goat-anti-rabbit-APC (Southern Biotech, Cat# 4050-11S; RRID:AB_2795959), mouse anti-human monoclonal antibodies against p75/NGFR (Becton-Dickinson, Cat# 40-1457) bound to phycoerythrin (PE), and EGFR (Fitzgerald, Acton, MA, Cat# 61R-E109BAF; RRID:AB_10808749) bound to FITC on ice in PBS/0.2% human serum for 30 min. After washing, we re-suspended cells in PBS/0.2% human serum containing 2 µg/mL 7-aminoactinomycin D (7-AAD) (Invitrogen, Cat# A1310). We carried out isotopic controls with irrelevant mouse IgG1-APC, mouse-IgG1-PE, and mouse-IgG1-FITC in parallel. Cells were FACS-sorted using a four-laser FACSDiva (Becton-Dickinson) to acquire alive SC sub-population (P75+/EGFR+/*P2ry14*+/7-AAD- and P75+/EGFR+/*P2ry14*-/7-AAD-). Three primary human PNs were FACS-sorted independently.

### Western blotting
Primary antibodies and dilutions were: Anti-Purinergic Receptor *P2ry14* rabbit (extracellular) (1:200, Sigma-Aldrich, Cat#: P0119; RRID:AB_1078921); GPR105 Polyclonal antibody rabbit (1:200, Thermo Fisher, Cat#: PA5-34087; RRID:AB_2551440); phospho PKA substrate (RRXS*/T*) (100G7E) rabbit (1:1000, Cell Signaling Technology, Cat#: 9624S; RRID:AB_331817).

## Sphere culture

We dissociated DRG from E12.5 embryos with 0.25% Trypsin (Thermo Fisher, Cat# 25200056) for 20 min at 37°C and obtained single-cell suspensions with narrow-bore pipettes and a 40 μm strainer (BD-Falcon), plating the cells in 24-well low attachment plates (Corning). The free-floating cells were cultured in serum-free medium with EGF and FGF as described (*Williams et al., 2008*). For passage, we dissociated spheres with 0.05% Trypsin (Thermo Fisher, Cat# 25300054) at 37°C for 5 min. For shRNA treatment and sphere counts, we plated SCP cells at low density to avoid sphere fusion (1000 cells/well in 24-well plates). In the *P2ry14* drug studies, we treated the cells with *P2ry14* inhibitor: *P2ry14* Antagonist Prodrug 7j hydrochloride (Axon Medchem, Cat# 1958) at concentrations of 30, 100, 300, and 500 nM. Three biological replicates produced similar results. Dose-response analysis confirmed that the optimal concentration of PPTN was 300 nM in this assay. For the shRNA experiments, we treated cells with three different lentiviral particles: shRNA control plasmid DNA (Sigma-Aldrich, Cat# SHC016-1EA), sh*P2ry14*(09) (Sigma-Aldrich, Cat# SHCL-NG-NM_133200; TRCN0000328609), sh*P2ry14*(84) (Sigma-Aldrich, Cat# SHCLNG-NM_133200; TRCN0000328684), sh*P2ry14*(64) (Sigma-Aldrich, Cat# SHCLNG-NM_133200; TRCN0000026664) at MOI = 10, 24 hr after plating. For rolipram experiment, we treated cells with 1 μM rolipram (Selleck, Cat# S1430). To passage, we centrifuged sphere cultures, treated with 0.05% Trypsin for 3 min, dissociated and plated at $2 \times 10^4$ cells/mL in 50% conditioned and 50% fresh medium. We counted secondary spheres after 14 days. For every cell line three biological replicates with three technical replicates were done. Of the three biological replicates, the best one was reported as a representative (n = 3). Spheres were counted with an inverted phase contrast microscope after 6 days of plating.

## Primary SC culture

E12.5 primary mouse SCs were isolated from DRG with neuronal contact in N2 medium with nerve growth factor, then removed from neurons and cultured in SC media (DMEM [Thermo Fisher, Cat# 11965118] + 10% FBS [Gemini Bio-Products, Cat# A87G02J] + β-heregulin peptide [R&D Systems, Cat# 396-HB-050] + forskolin [Cayman Chemical, Cat# 11018]) for one to three passages as described (*Kim et al., 1995*).

## Direct cAMP ELISA

Primary SCs were plated on poly-L-lysine coated six-well plate (~750,000 cells/well) in DMEM + 10% FBS + β-HRG + forskolin. After cells were confluent, cells were starved with serum-free N2 media with N2 supplement and left incubating for 16 hr. Cells were pre-incubated with MEK inhibitor for 2 hr prior to stimulation. Stimulation was carried out for 2 min and cAMP measurement was done according to manufacturer's protocol (Direct cAMP ELISA kit by Enzo Life Sciences, Cat# ADI-900-066) using the acetylated format. An aliquot prior to cAMP measurements was set aside for protein quantification using the Bio-Rad protein assay kit. For every cell line two biological replicates with three technical replicates were done. Of the two biological replicates the best one was reported as a representative (n = 3 biological replicates, 3 technical replicates).

## Immunostaining

For frozen sections, OCT was removed by incubation with 1× PBS. We permeabilized cells in ice-cold MeOH for 10 min, followed by incubation in normal donkey serum (Jackson ImmunoResearch, Cat# 017-000-121; RRID:AB_2337258) and 0.3% Triton-X100 (Sigma-Aldrich, Cat# X100). Ki67 (1:200, Cell Signaling Technologies, Cat# 12202S; RRID:AB_2620142), anti-CNPAse (1:250, Sigma-Aldrich Millipore, Cat# AB9342; RRID:AB_569543); β-galactosidase polyclonal antibody (1:1000, Thermo Fisher, Cat#: A-11132; RRID:AB_221539). All secondary antibodies were donkey anti-Rat/Rabbit/Goat from Jackson ImmunoResearch, reconstituted in 50% glycerol and used at 1:250 dilution. To visualize nuclei, sections were stained with DAPI for 10 min, washed with PBS and mounted in FluoromountG (Electron Microscopy Sciences, Hatfield, PA). Images were acquired with ImageJ Acquisition software using a fluorescence microscope (Axiovert 200M) with 10×/0.4 or 40×/0.6 objectives (Carl Zeiss, Inc), or with NIS-Elements software using confocal microscopy (Nikon).

## Electron microscopy

Mice were perfusion fixed with 4% paraformaldehyde and 2.5% glutaraldehyde in 0.1 M phosphate buffer at 7.4 pH. Saphenous nerve was dissected out and postfixed overnight, then transferred to 0.175 mol/L cacodylate buffer, osmicated, dehydrated, and embedded in Embed 812 (Ladd Research Industries). Ultrathin sections were stained in uranyl acetate and lead citrate and viewed on a Hitachi H-7600 microscope.

## Mouse dissection and quantification of neurofibroma number and size

To quantify neurofibroma number and size, we perfused mice and used a Leica dissecting microscope to dissect the spinal cord with attached DRG and nerve roots at the ages of 4 and 7 months, as previously described (*Wu et al., 2016*). A neurofibroma was defined as a mass surrounding the DRG or nerve roots, with a diameter greater than 1 mm, measured perpendicular to DRG/nerve roots. Neurofibroma diameter for each mouse was measured with ImageJ.

## Osmotic pump drug study

Four-month-old neurofibroma mice $Nf1^{fl/fl};Dhh^{Cre}$ were treated with the *P2ry14* inhibitor (PPTN HCl) (TOCRIS: Cat. 4862). Osmotic minipumps were loaded with the *P2ry14* inhibitor drug according to manufacturer's instructions with PPTN in saline solution (containing DMSO) or with saline containing equivalent amounts of the vehicle control (DMSO). Osmotic minipumps released the inhibitor hourly for 14 days for a daily dose of 4.55 mg/kg. At day 14, mice were perfusion fixed with 4% paraformaldehyde and used a Leica dissecting microscope to dissect the tissue.

## RT-PCR

We isolated total RNA from WT and Nf1 spheres treated with sh*P2ry14* using the RNeasy Plus Micro-Kit (QIAGEN, Cat# 74034) and made cDNA using the High-Capacity Reverse Transcription Kit (Thermo Fisher, Cat# 4368813). We conducted RT-PCR as described in *Williams et al., 2008*. Primer sequence: *P2ry14* forward: 5'-AGCAGATCATTCCCGTGTTGT-3'. *P2ry14* reverse: 5'-TCTCAAGAACAT AGTGGTGGCT-3'.

## Genotyping

Power SYBR Green PCR Master Mix (Thermo Fisher, Cat# 4368702) was used for genotyping. We genotyped mice as described by *Meister et al., 2014*. Genotyping primers: β-galactosidase sense: 5'-AGAAGGCACATGGCTGAATATCGA-3'. *P2ry14* forward: (5'-AGCTGCCGGACGAAGGAGACCCTG CTC-3'). *P2ry14* reverse: 5'-GGTTTTGGAAACCTCTAGGTCATTCTG- 3' (*Meister et al., 2014*).

## Statistics

Statistical parameters, including the type of tests, number of samples (n), descriptive statistics, and significance are reported in the figures and figure legends. Two-group comparisons used Student's t-tests. When single agents were tested at different concentration in a single cell type, we used a one-way ANOVA with a Dunnett's multiple comparisons test. When multiple genotypes were analyzed in a single experiment, we used a two-way ANOVA with multiple comparisons, without matching, and correction with the Holm-Sidak test. Mann-Whitney test was used for comparisons between genotypes for tissue widths and neurofibroma incidence (GraphPad Prism V9). All data unless otherwise stated is represented as average ± SD, and was analyzed in GraphPad Prism 7.

## Acknowledgements

We thank Takeda Cambridge Limited and Dr Johannes Grosse for providing the P2RYR14 knock-in mouse. For excellent technical assistance, we thank Mark Jackson and Lindsey E Aschbacher-Smith. Funding: JP-C was supported by NIH-T32-NS007453 and a Children's Tumor Foundation Young Investigator Award. This work was supported by grants NIH-R01-NS28840 and NIH-R37-NS083580 to NR.

# Additional information

## Funding

| Funder | Grant reference number | Author |
|---|---|---|
| National Institutes of Health | T32-NS007453 | Jennifer Patritti Cram |
| Children's Tumor Foundation Young Investigator Award | 2020-01-006 | Jennifer Patritti Cram |
| National Institutes of Health | NIH-R01-NS28840 | Nancy Ratner |
| National Institutes of Health | NIH-R37-NS083580 | Nancy Ratner |

The funders had no role in study design, data collection and interpretation, or the decision to submit the work for publication.

## Author contributions

Jennifer Patritti Cram, Conceptualization, Formal analysis, Funding acquisition, Investigation, Methodology, Validation, Visualization, Writing - original draft, Writing – review and editing; Jianqiang Wu, Formal analysis, Investigation, Methodology, Validation, Visualization, Writing – review and editing; Robert A Coover, Methodology, Validation, Writing – review and editing; Tilat A Rizvi, Katherine E Chaney, Methodology; Ramya Ravindran, Formal analysis, Methodology; Jose A Cancelas, Formal analysis, Methodology, Visualization, Writing – review and editing; Robert J Spinner, Resources; Nancy Ratner, Conceptualization, Formal analysis, Funding acquisition, Resources, Supervision, Visualization, Writing - original draft, Writing – review and editing

## Author ORCIDs

Jennifer Patritti Cram ⓘ http://orcid.org/0000-0001-5971-0849
Jianqiang Wu ⓘ http://orcid.org/0000-0002-4239-5659
Nancy Ratner ⓘ http://orcid.org/0000-0001-5030-9354

## Ethics

This study was performed in strict accordance with the recommendations in the Guide for the Care and Use of Laboratory Animals of the National Institutes of Health. All of the animals were handled according to approved institutional animal care and use committee (IACUC) protocols (#2018-0103 expiration 01-2022) of Cincinnati Children's Hospital. The protocol was approved by the Committee on the Ethics of Animal Experiments of the Cincinnati Children's Hospital.

## Decision letter and Author response

Decision letter https://doi.org/10.7554/eLife.73511.sa1
Author response https://doi.org/10.7554/eLife.73511.sa2

# Additional files

## Supplementary files
• Transparent reporting form
• Source data 1. Non-annotated original figures.
• Source data 2. Annotated original figures.

## Data availability

The data sets and original figures generated during this study will be available at Synapse Project (https://www.synapse.org/).

The following dataset was generated:

| Author(s) | Year | Dataset title | Dataset URL | Database and Identifier |
|---|---|---|---|---|
| Patritti Cram J | 2022 | Understanding the role of purinergic signaling on tumor formation in a mouse model of Nf1 | https://www.synapse.org/#!Synapse:syn27752640 | Synapse, 10.7303/syn27752640 |

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
