## [Editor Report]

This study explores a role for the purinergic receptor P2RY14 and cAMP signaling in Schwann cell precursor self-renewal and neurofibroma development. Importantly, the authors show that genetic and chemical inhibition of P2RY14 inhibits Schwann cell precursor self-renewal in vitro and suppresses neurofibroma development in vivo. The authors also report that these effects are mediated by an increase in cAMP signaling.

---

## [Decision Letter]

**Decision letter after peer review:**

[Editors’ note: the authors submitted for reconsideration following the decision after peer review. What follows is the decision letter after the first round of review.]

Thank you for submitting the paper "Purinergic receptor P2RY14 cAMP signaling regulates EGFR-driven Schwann cell precursor self-renewal and nerve tumor initiation in neurofibromatosis" for consideration by *eLife*. Your article has been reviewed by 2 peer reviewers, and the evaluation has been overseen by a Reviewing Editor and a Senior Editor. The reviewers have opted to remain anonymous.

We are sorry to say that, after consultation with the reviewers, we have decided that this work will not be considered further for publication by *eLife*.

Both reviewers recognized the significant work presented in this manuscript and felt that the connection between P2RY14 loss, cAMP pathway activation, SCP self-renewal and neurofibroma development is well supported. However, the major observation describing the relationship between P2RY14 and EGFR signaling was supported by experiments performed in a number of different cell systems without adequate demonstration that the systems provide insight into the same biology. The connections between human and murine cells was also not always consistent. For these reasons, the reviewers remained skeptical that this relationship is as described in the manuscript. Since considerable work would be necessary to perform a significant number of new experiments in different systems, we are returning this manuscript to you so that you might consider other options for the work.

*Reviewer #1:*

The NF1 tumor suppressor regulates the Ras pathway but it has also been implicated in cAMP signaling. In this study the authors first sought to identify receptor(s), impinging on the cAMP pathway, that might be involved in neurofibroma development. Notably, they found that the purinergic receptor P2RY14 is enriched in Schwann cell precursors, the cell of origin for neurofibromas. They then demonstrate that (1) genetic and chemical suppression of P2RY14 results in enhanced cAMP signaling, (2) this increased cAMP signaling inhibits the enhanced self-renewal of NF1-deficient Schwann cell precursors, and (3) genetic P2RY14 ablation suppresses neurofibroma initiation in a GEMM model.

Strengths: A major strength of this manuscript are the impressive in vivo phenotypes. Specifically, the authors convincingly demonstrate that genetic P2RY14 ablation reduces neurofibroma development in a well-characterized mouse (GEMM) model. Another strength lies in the in vitro studies examining self-renewal of Schwann cell precursors (SCPs), which suggest that tumor suppression mediated by P2RY14-loss occurs by inhibiting tumor initiation due to an increase in cAMP signaling. Together these findings reveal a new and important pathway that contributes to neurofibroma development. Moreover, this study suggests that P2RY14 may represent a potential therapeutic target in these tumors.

Weaknesses: Because NF1 regulates the Ras pathway and converges on EGFR signaling in SCPs, the authors also examine whether P2RY14 impinges on these pathways. They do show that P2RY14 inhibition attenuates either AKT, ERK and/or EGFR in various systems (mouse SCPs, human SCPs, and human MPNST cells) in vitro. However, the relative importance of these pathways downstream of P2RY14 is not clear as presented. This is in part because not all 3 events (pERK, pAKT, pEGFR) are evaluated in all settings, and effects appear to be minimal in vivo (and/or not examined). This may be due to the fact that the phenotype appears to be primarily related to SCP self-renewal and tumor initiation, and therefore signaling defects may be difficult to capture in this context.

While the connection between P2RY14 loss, cAMP pathway activation, SCP self-renewal and neurofibroma development is well supported, the connection between P2RY14 loss and EGFR, pERK, pAKT is less clear in the current version of this manuscript. For example, in immortalized human Schwann cells, P2RY14 suppression attenuates pEGFR and pAKT, but has no effect on pERK. By contrast, only ERK is examined in mouse SCPs and it is only modestly suppressed, whereas both ERK and AKT are both attenuated in MPNSTs. Finally, only pERK is examined in vivo, and it is not convincingly suppressed by P2RY14 loss.

If the authors wish to claim that P2RY14 loss impacts EGFR signaling, it would be important to measure the same signals in all settings. For example, can they examine pEGFR, pAKT and pERK in the mouse cells used in Figure 2F? If not I would suggest deleting this panel altogether, because it raises doubt about later figures.

Similarly, can the authors also examine pEGFR and pAKT in vivo in 5I? (Although, unfortunately the pERK is not very convincing). Ultimately, if the authors really believe that tumor suppression is due to direct suppression of EGFR activation/phosphorylation, pEGFR should be inhibited in all settings (while defects in ERK and AKT may vary). Of course, it is possible that signaling defects might not be easily captured in vivo, especially if the biological defect is in tumor initiation and therefore may not be present in many cells in established tumors. If that is the case then it may be better not to show the minimal effects on pERK at all in vivo in panel 5I, especially given the very dramatic effects on PKA shown in panel 5H.

The authors claim that P2RY14 and EGFR physically interact, however additional negative controls are required to support this conclusion. If they wish to claim that there is a physical association, they must use a GFP-expressing control and then IP with anti-GFP rather than performing a mock immunoprecipitation (given the sticky nature of GFP). Nevertheless, proof of a direct interaction is not necessary for the concept of feedback interference with EGFR signaling (depending on supportive signaling data described above).

Overall, I would suggest that the authors either (1) strengthen the P2Y14-EGFR connection, (2) only include the experiments shown in Figure 3 and temper conclusions suggesting the involvement of EGFR suppression in this phenotype (eg "suppression of EGFR signaling may also contribute"), or (3) remove the discussion of cross-talk suppression altogether.

*Reviewer #2:*

Using patient-derived neurofibroma samples, the authors first showed that compared to p75+EGFR- neurofibroma cells, P2RY14 was expressed at higher levels in p75+EGFR+ cells, which was also correlated with higher neurosphere-forming abilities in vitro. Secondly, the authors showed that inhibition of P2RY14 by either a specific inhibitor, PPTN, or shRNAs for P2RY14, inhibited neurosphere-forming activities of Nf1-/- cultured mouse Schwann cell precursor (SCP) cells, but exhibited little effect on Nf1+/+ cultures. Thirdly, the authors showed that immortalized human Schwann cells (iHSCs) without NF1 (NF1-/-) exhibited increased growth advantage compared with NF1+/+ counterparts. Using P2RY14-specific UDP agonist, P2RY14 inhibitor and IBMX (an inhibitor of PDEs), the authors showed that activation versus inhibition of P2RY14 activities are correlated with a decrease versus an increase of cAMP signaling based on pCREB expression. In NF1-/- iHSCs, P2RY14 inhibition reduced phosphorylation of EGFR and shortened the duration of p-AKT signaling. Finally, germline knockout of P2RY14 reduced neurofibroma burden and extended life span of the Dhh-cre driven NF1-associated neurofibroma model. However, treatment of Rolipram, which inhibits PDE4 and blocks degradation of cAMP, increased cAMP and decreased proliferation in neurofibromas, but failed to significantly alter tumor volume in the Dhh-cre driven neurofibroma model. This study presents a lot of data derived from different types of in vitro and in vivo systems, including from both humans and mouse models. However, it is not clear why different in vitro or in vivo models were used to investigate the hypothesis. The conclusions from human and mouse systems regarding the role of P2RY14 in neurofibroma initiation versus progression, are not necessarily consistent with each other. The relationship between P2RY14 and EGFR signaling is not fully established in neurofibroma cells.

The paper needs to be restructured, focuses on one or few points of the paper and perform additional experiments to justify the conclusions.

It would strengthen the manuscript if the following questions are addressed.

1. Studies using patient-derived neurofibroma cells suggest a role of P2RY14 in promoting neurofibroma growth (Figure 1). However, mouse studies using Rolipram treatment have little effect on neurofibroma volume, despite increasing cAMP signaling (Figure 6). How can these human and mouse studies be reconciled?

2. It is not clear why inhibition of P2RY14 in cultured mouse SCP cells only inhibits the neurosphere-forming abilities of NF1-/- cells, but not NF1+/+ cells, though cAMP signaling was similarly increased (Figure 2G). Do these data argue against the notion that P2PY14 signals through the cAMP signaling pathway.

3. In Figure 3, the relationship between P2RY14, cAMP and EGFR pathways is at best correlative, but lacks direct evidence to establish a causal relationship. Further, it is not clear why malignant peripheral nerve sheath tumor cell lines were used in these studies (Figure 3E-G), which may not be relevant to neurofibroma cells.

4. The observation that germline P2PY14 knockout inhibits neurofibroma formation in the Dhh-cre driven mouse model is interesting (Figure 4 and 5). However, it is not clear how this inhibition is related to cAMP and/or EGFR signaling pathways in vivo?

---

## [Author Response]

[Editors’ note: The authors appealed the original decision. What follows is the authors’ response to the first round of review.]

Reviewer #1:The NF1 tumor suppressor regulates the Ras pathway but it has also been implicated in cAMP signaling. In this study the authors first sought to identify receptor(s), impinging on the cAMP pathway, that might be involved in neurofibroma development. Notably, they found that the purinergic receptor P2RY14 is enriched in Schwann cell precursors, the cell of origin for neurofibromas. They then demonstrate that (1) genetic and chemical suppression of P2RY14 results in enhanced cAMP signaling, (2) this increased cAMP signaling inhibits the enhanced self-renewal of NF1-deficient Schwann cell precursors, and (3) genetic P2RY14 ablation suppresses neurofibroma initiation in a GEMM model.Strengths: A major strength of this manuscript are the impressive in vivo phenotypes. Specifically, the authors convincingly demonstrate that genetic P2RY14 ablation reduces neurofibroma development in a well-characterized mouse (GEMM) model. Another strength lies in the in vitro studies examining self-renewal of Schwann cell precursors (SCPs), which suggest that tumor suppression mediated by P2RY14-loss occurs by inhibiting tumor initiation due to an increase in cAMP signaling. Together these findings reveal a new and important pathway that contributes to neurofibroma development. Moreover, this study suggests that P2RY14 may represent a potential therapeutic target in these tumors.Weaknesses: Because NF1 regulates the Ras pathway and converges on EGFR signaling in SCPs, the authors also examine whether P2RY14 impinges on these pathways. They do show that P2RY14 inhibition attenuates either AKT, ERK and/or EGFR in various systems (mouse SCPs, human SCPs, and human MPNST cells) in vitro. However, the relative importance of these pathways downstream of P2RY14 is not clear as presented. This is in part because not all 3 events (pERK, pAKT, pEGFR) are evaluated in all settings, and effects appear to be minimal in vivo (and/or not examined). This may be due to the fact that the phenotype appears to be primarily related to SCP self-renewal and tumor initiation, and therefore signaling defects may be difficult to capture in this context.While the connection between P2RY14 loss, cAMP pathway activation, SCP self-renewal and neurofibroma development is well supported, the connection between P2RY14 loss and EGFR, pERK, pAKT is less clear in the current version of this manuscript. For example, in immortalized human Schwann cells, P2RY14 suppression attenuates pEGFR and pAKT, but has no effect on pERK. By contrast, only ERK is examined in mouse SCPs and it is only modestly suppressed, whereas both ERK and AKT are both attenuated in MPNSTs. Finally, only pERK is examined in vivo, and it is not convincingly suppressed by P2RY14 loss.

All removed.

If the authors wish to claim that P2RY14 loss impacts EGFR signaling, it would be important to measure the same signals in all settings. For example, can they examine pEGFR, pAKT and pERK in the mouse cells used in Figure 2F? If not I would suggest deleting this panel altogether, because it raises doubt about later figures.

All removed.

Similarly, can the authors also examine pEGFR and pAKT in vivo in 5I? (Although, unfortunately the pERK is not very convincing). Ultimately, if the authors really believe that tumor suppression is due to direct suppression of EGFR activation/phosphorylation, pEGFR should be inhibited in all settings (while defects in ERK and AKT may vary). Of course, it is possible that signaling defects might not be easily captured in vivo, especially if the biological defect is in tumor initiation and therefore may not be present in many cells in established tumors. If that is the case then it may be better not to show the minimal effects on pERK at all in vivo in panel 5I, especially given the very dramatic effects on PKA shown in panel 5H.

Done; Figure 5I was deleted.

The authors claim that P2RY14 and EGFR physically interact, however additional negative controls are required to support this conclusion. If they wish to claim that there is a physical association, they must use a GFP-expressing control and then IP with anti-GFP rather than performing a mock immunoprecipitation (given the sticky nature of GFP). Nevertheless, proof of a direct interaction is not necessary for the concept of feedback interference with EGFR signaling (depending on supportive signaling data described above).

We agree and removed this data.

Overall, I would suggest that the authors either (1) strengthen the P2Y14-EGFR connection, (2) only include the experiments shown in Figure 3 and temper conclusions suggesting the involvement of EGFR suppression in this phenotype (eg "suppression of EGFR signaling may also contribute"), or (3) remove the discussion of cross-talk suppression altogether.

We agree. We removed all discussion of cross-talk and will consider a separate, strengthened manuscript on cross-talk.

Reviewer #2:Using patient-derived neurofibroma samples, the authors first showed that compared to p75+EGFR- neurofibroma cells, P2RY14 was expressed at higher levels in p75+EGFR+ cells, which was also correlated with higher neurosphere-forming abilities in vitro. Secondly, the authors showed that inhibition of P2RY14 by either a specific inhibitor, PPTN, or shRNAs for P2RY14, inhibited neurosphere-forming activities of Nf1-/- cultured mouse Schwann cell precursor (SCP) cells, but exhibited little effect on Nf1+/+ cultures. Thirdly, the authors showed that immortalized human Schwann cells (iHSCs) without NF1 (NF1-/-) exhibited increased growth advantage compared with NF1+/+ counterparts. Using P2RY14-specific UDP agonist, P2RY14 inhibitor and IBMX (an inhibitor of PDEs), the authors showed that activation versus inhibition of P2RY14 activities are correlated with a decrease versus an increase of cAMP signaling based on pCREB expression. In NF1-/- iHSCs, P2RY14 inhibition reduced phosphorylation of EGFR and shortened the duration of p-AKT signaling. Finally, germline knockout of P2RY14 reduced neurofibroma burden and extended life span of the Dhh-cre driven NF1-associated neurofibroma model. However, treatment of Rolipram, which inhibits PDE4 and blocks degradation of cAMP, increased cAMP and decreased proliferation in neurofibromas, but failed to significantly alter tumor volume in the Dhh-cre driven neurofibroma model. This study presents a lot of data derived from different types of in vitro and in vivo systems, including from both humans and mouse models. However, it is not clear why different in vitro or in vivo models were used to investigate the hypothesis. The conclusions from human and mouse systems regarding the role of P2RY14 in neurofibroma initiation versus progression, are not necessarily consistent with each other. The relationship between P2RY14 and EGFR signaling is not fully established in neurofibroma cells.The paper needs to be restructured, focuses on one or few points of the paper and perform additional experiments to justify the conclusions.

We agree completely and extensively revised the manuscript.

It would strengthen the manuscript if the following questions are addressed.1. Studies using patient-derived neurofibroma cells suggest a role of P2RY14 in promoting neurofibroma growth (Figure 1). However, mouse studies using Rolipram treatment have little effect on neurofibroma volume, despite increasing cAMP signaling (Figure 6). How can these human and mouse studies be reconciled?

We removed the Rolipram volume data because numbers of treated mice were insufficient for statistical analysis. Experiments would require a year to generate and treat the 10-17 additional mice for accurate volumetric analysis. We replaced this with a statement that more work will be needed to define long term effects. In addition, to strengthen our argument we added a new study using a direct and specific P2RY14 antagonist, which confirms and extends the Rolipram data showing effects on cell proliferation.

2. It is not clear why inhibition of P2RY14 in cultured mouse SCP cells only inhibits the neurosphere-forming abilities of NF1-/- cells, but not NF1+/+ cells, though cAMP signaling was similarly increased (Figure 2G). Do these data argue against the notion that P2PY14 signals through the cAMP signaling pathway.

Data in Figure 2 shows that PPTN or shRNA targeting P2RY14 have effects on both wild type and on Nf1 mutant SCP spheres. The key result is that effects are significantly more in the mutant versus wild type cells. This differential effect is consistent with de-regulated cell signaling in Nf1 mutant cells. For example, both WT and mutant cells use P-ERK, yet MAPK inhibition shows more effect on mutant versus wild type cells in vivo. We clarified this in the text in lines 182-184.

3. In Figure 3, the relationship between P2RY14, cAMP and EGFR pathways is at best correlative, but lacks direct evidence to establish a causal relationship. Further, it is not clear why malignant peripheral nerve sheath tumor cell lines were used in these studies (Figure 3E-G), which may not be relevant to neurofibroma cells.

We agree and so removed this data.

4. The observation that germline P2PY14 knockout inhibits neurofibroma formation in the Dhh-cre driven mouse model is interesting (Figure 4 and 5). However, it is not clear how this inhibition is related to cAMP and/or EGFR signaling pathways in vivo?

Unfortunately, we cannot directly address this question in vivo. However, given that blockade of either EGFR or P2RY14 in vivo and in SCP in vitro reduces tumor initiation provides strong correlation, but does not demonstrate causality for the in vivo effect, which is provided by the in vitro experiments. For this reason, we add data to show in vivo that raising cAMP levels by rolipram or by P2RY14 antagonism similarly affect neurofibroma cell proliferation.